# An Experimental Study of the Failure Mode of ZnO Varistors Under Multiple Lightning Strokes

**Chunlong Zhang** [1,2,3,4], **Hongyan Xing** [1,2,3,*], **Pengfei Li** [3,4], **Chunying Li** [3,4], **Dongbo Lv** [3,4] **and Shaojie Yang** [4]

[1] Collaborative Innovation Center on Forecast and Evaluation of Meteorological Disasters, Nanjing University of Information Science & Technology, Nanjing 210044, China; 20151114051@nuist.edu.cn

[2] Jiangsu Key Laboratory of Meteorological Observation and Information Processing, Nanjing University of Information Science & Technology, Nanjing 210044, China

[3] School of Atmospheric Physics, Nanjing University of Information Science & Technology, Nanjing 210044, China; 20131214195@nuist.edu.cn (P.L.); 20156282032@nuist.edu.cn (C.L.); 20156282031@nuist.edu.cn (D.L.)

[4] Meteorological Disaster Prevention Technology Center of Heilongjiang Province, Harbin, 150030, China; sjyang@grmc.gov.cn

[*] Correspondence: xinghy@nuist.edu.cn; Tel.: +86-025-58731370

**Abstract:** In this study, in order to explore the failure mode of ZnO varistors under multiple lightning strokes, a five-pulse 8/20 μs nominal lightning current with pulse intervals of 50 ms was applied to ZnO varistors. Scanning electron microscopy (SEM) and X-ray diffractometry (XRD) were used to analyze the microstructure of the material. The failure processes of ZnO varistors caused by multiple lightning impulse currents were described. The performance changes of ZnO varistors after multiple lightning impulses were analyzed from both macro and micro perspectives. According to the results of this study's experiments, the macroscopic failure mode of ZnO varistors after multiple lightning impulses involved the rapid deterioration of the electrical parameters with the increase of the number of impulse groups, until destruction occurred by side-corner cracking. The microstructural examination indicated that, after the multiple lightning strokes, the proportion of Bi in the crystal phases was altered, the grain size of the ZnO varistors became smaller, and the white intergranular phase (Bi-rich grain boundary layer) increased significantly. The failure mechanism was thermal damage and grain boundary structure damage caused by temperature gradient thermal stress, generated by multiple lightning currents.

**Keywords:** failure mode; impulse current; microstructure; multiple lightning; ZnO varistors

## 1. Introduction

The installation of surge protective devices is one of the most economical and effective means to avoid or reduce damage caused by lightning impulses in power distribution systems. The core component of a surge protective device is a ZnO varistor, which has many advantages, including a good nonlinear property, small normal leakage current, low level of residual pressure, and no follow current. ZnO varistors have been widely applied for the protection of electronic and electrical equipment from lightning [1,2]. However, after a natural lightning strike, ZnO varistors often have varying degrees of failure.

An impulse test is the most direct means to study the performance of ZnO varistors. Currently, an 8/20 μs single pulse waveform had been adopted for such tests [3,4]. A host of studies have been presented regarding the performance changes that occur in ZnO varistors under single pulse lightning impulses [5–8]. However, modern lightning observations and artificially triggered lightning

acquisition data have shown that two-thirds of the lightning events in natural settings are a multi-pulse process. The statistics show that nearly 70% of cloud-to-ground lightning strikes involve from 2 and up to 20 strikes, with an average number of between 3 and 5, and a time duration between strikes of 15 ms to 150 ms. There is a significant difference between the single-pulse lightning waveform and the multiple lightning strikes of natural lightning [9,10], and the total time and energy of a multi-pulse are several times that of a single pulse. When a low-voltage distribution line is struck by a lightning, the lightning current flowing through the surge protective device generates heat in the ZnO varistor due to power loss, causing the body to heat up. When a multi-pulse lightning current and single-pulse lightning current flow through a ZnO varistor, there is a huge difference in duration and energy absorption, which inevitably leads to a difference in the temperature increase of the ZnO varistor and the final electrical performance parameters. Therefore, lightning single pulse test methods have been unable to properly simulate the damage caused by lightning [11]. Darveniza et al. [12] pointed out that multiple lightning impulse currents could potentially cause severe damage to lightning protection devices protecting the equipment in power distribution systems. Previous studies have shown that six-pulse lightning current impulses have resulted in serious damage to lightning protection components, and confident conclusions cannot currently be drawn regarding their impact according to the lightning protection test standard. Lee et al. [13,14] found that ZnO varistors degraded when subjected to multiple impulse currents, and their life mainly depended on the amplitude of the lightning surge. Haryono et al. [15,16] analyzed the damage effects to ZnO varistors undergoing multiple lightning impulse currents from the perspective of energy absorption. At the present time, the research regarding the performance of ZnO varistors undergoing multiple lightning impulse current has been mainly based on the analysis of the macroscopic electrical properties [17–19]. Qingheng Chen et al. [20] used the simulation network model to analyze the current, temperature, and thermal stress in zinc oxide varistors. The results showed that reducing the average size of the ZnO grains can significantly reduce the temperature difference inside the ZnO varistor. Thermal stress increases the impact energy absorption capacity of zinc oxide varistors. Pengfei Li et al. [21] analyzed the damage form of metal oxide under multiple lightning impulses and found that it was mainly the edge that was cracking. However, few research studies have been conducted to investigate the damage failure mechanisms under multi-pulse continuous impulses, especially regarding the changes in microstructures. Tsuboi et al. [22] believed that damage could occur in the internal components of ZnO when subjected to multiple lightning impulses, resulting in the failure of the ZnO varistor, which provided a theoretical basis for the modeling of this study.

Therefore, in this study, a five-pulse current with a time interval of 50 ms was applied to the samples. The actual lightning strike to the ZnO varistors in practical applications was simulated as realistically as possible, and the macroscopic damage form and static parameter variation characteristics were monitored. At the same time, the ZnO varistors were analyzed before and after the impulse by scanning electron microscopy (SEM) and X-ray diffractometry (XRD). It was expected that we would investigate the failure mode of the ZnO varistors during the natural lightning strikes from the perspective of macroscopic and microscopic bonding. Indeed, some experimental data regarding the performance of ZnO varistors under multiple lightning impulse currents are shown here. The failure modes of the ZnO varistors undergoing multiple lightning impulse currents are discussed on the basis of the results of thermal effect. This is believed to be particularly important for improving lightning protection and safety performances of ZnO varistors.

## 2. Experiment

### 2.1. Impulse Test and Waveform

The multiple lightning impulse equipment used in this experiment was a 20-pulse lightning impulse current generator, as shown in Figure 1. Multi-channel discharge technology was adopted to simulate the multiple lightning stroke processes. The waveform generator circuit diagram is shown

in Figure 2, in which C is the capacitor, G is the impact gap, Rs is the resistor, and Ls is the inductor. With the impulse test end as the axis, there are 10 trigger channels at each end. When the primary current is fully triggered, 20 high-voltage pulses can be generated, and the time interval can be changed from 1 ms to 999 ms.

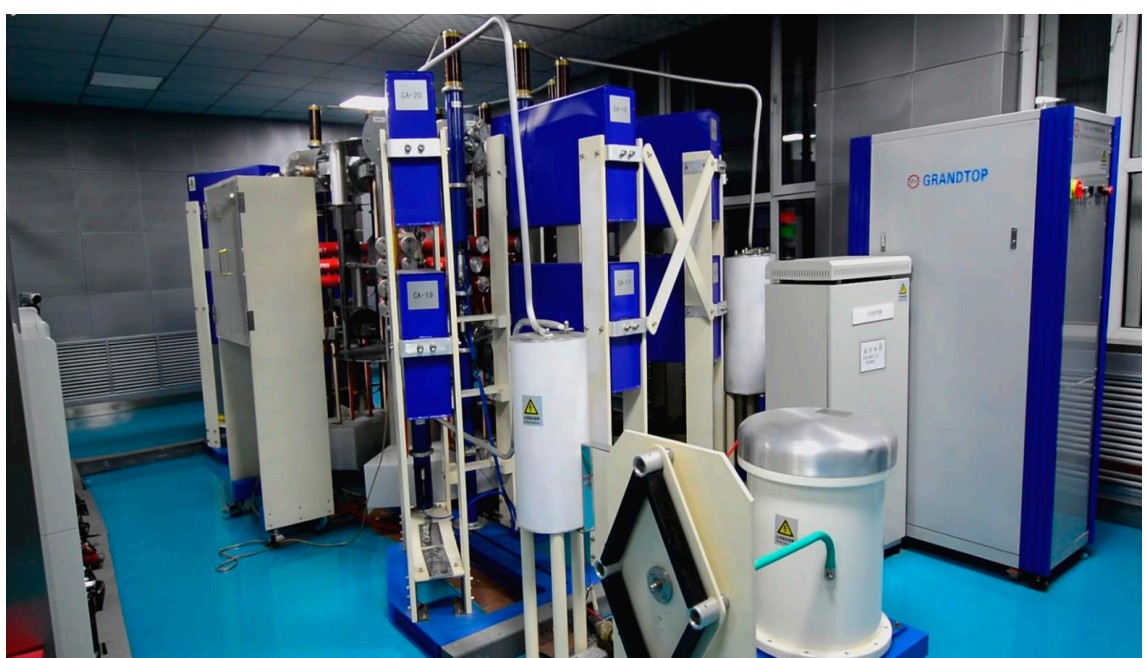

**Figure 1.** 20-pulse lightning impulse current generator.

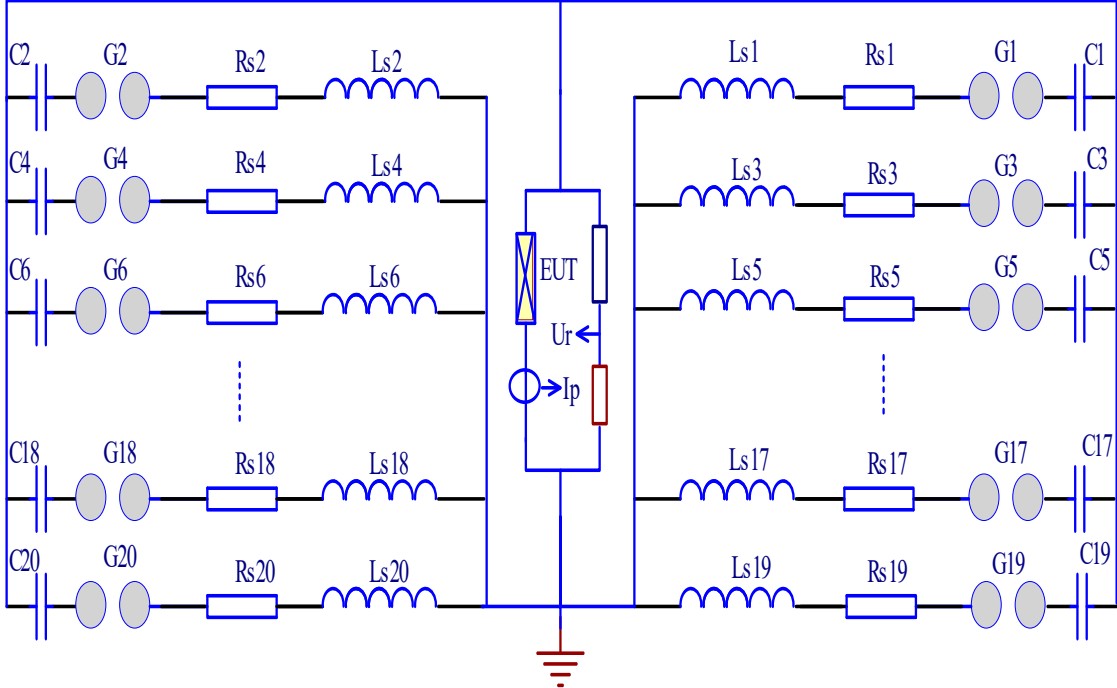

**Figure 2.** Multiple lightning impulse current generator circuit diagram.

In this study, Section 1 of IEC 62305-1 Lightning Protection: General Rule [23] provides a special definition for multiple lightning. Lightning with an average of three to four impulses, and intervals of approximately 50 ms was defined as multiple lightning. Therefore, the multiple lightning strikes in this test were represented by a group of multiple impulse currents, which included five consecutive

impulse currents, each of which had 8/20 μs waveforms. The time between two consecutive pulses was 50 ms, and the pulse amplitudes were the nominal discharge currents of the selected ZnO varistors. The waveform diagram is shown in Figure 3, where the five yellow vertical lines represent the five pulses, and the following 8/20 μs waveform denotes the full waveform figure of a single pulse.

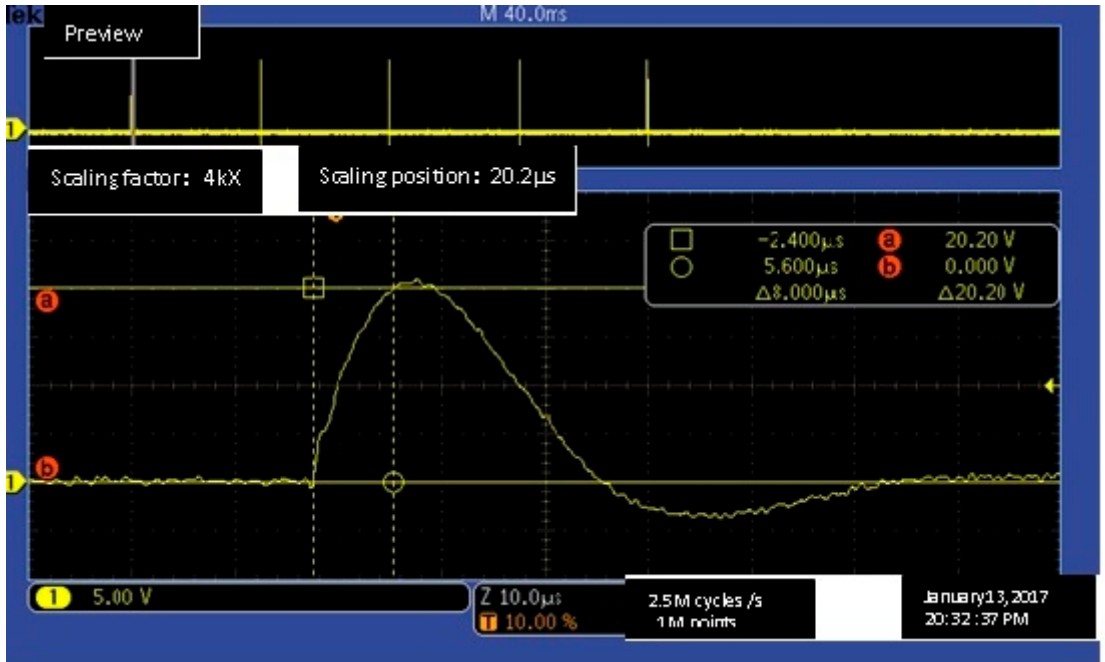

**Figure 3.** Five-pulse lightning waveform diagram.

## 2.2. Sample Preparation

In this study, the various ZnO varistors that were used were provided by the same manufacturer. The basic parameters of these samples were nominal discharge current $I_n$ = 20 kA and the maximum continuous operation voltage $U_c$ = 385 V, and the static parameters (varistor voltage and leakage current) were as shown in Table 1. The varistor voltage is a voltage corresponding to a current of the varistor of 1 mA and is a standard of a voltage whose current rapidly rises with voltage, expressed by $U_{1mA}$. The leakage current refers to the current flowing through the varistor at a specified temperature and maximum DC voltage, generally expressed by $I_{ie}$. Under the premise of approaching $U_{1mA}$, the smaller the $I_{ie}$ was, the better the voltage limiting the performance of the varistor would be. In this study, seven ZnO varistor blocks, with the majority approaching the static parameters, were selected as the samples and were denoted as A1 to A7.

**Table 1.** Changes in the electrical parameters before and after the impulse in the ZnO varistors.

| No. | Initial $U_{1mA}$ (V) | Initial $I_{ie}$ (μA) | $U_{1mA}$ When Failure (V) | $I_{ie}$ When Failure (μA) | $T_{max}$ When Failure (°C) | Groups of Impulse |
|-----|-----------------------|-----------------------|----------------------------|----------------------------|-----------------------------|-------------------|
| A1 | 690 | 0 | 602 | 8.2 | 224 | 17 |
| A2 | 689 | 0 | 600 | 8.1 | 225 | 16 |
| A3 | 690 | 0 | 604 | 7.5 | 222 | 15 |
| A4 | 691 | 0.1 | 615 | 9.4 | 221 | 16 |
| A5 | 688 | 0.1 | 605 | 8.6 | 216 | 16 |
| A6 | 690 | 0.1 | 603 | 8.5 | 228 | 15 |
| A7 | 689 | 0.1 | 602 | 7.8 | 218 | 16 |

## 2.3. Experimental Procedure

The flowchart of the experimental procedure is shown in Figure 4.

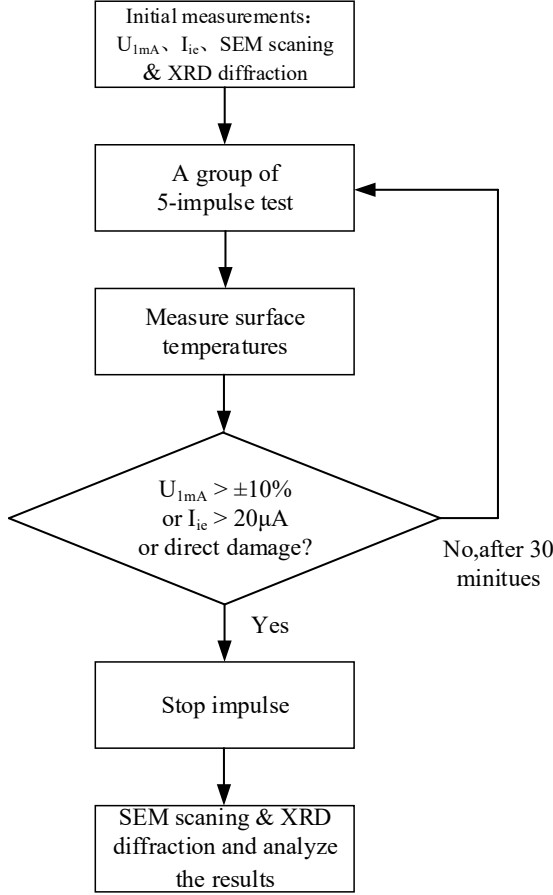

**Figure 4.** Flowchart of the experimental procedure.

(1) Initial measurements: The samples are characterized with the $U_{1mA}$, $I_{ie}$, and by photographs taken at the beginning of the test. We used scanning electron microscopy (SEM) and an X-ray diffractometer (XRD) on the ZnO varistor blocks before the impulse test.

(2) Impulse test: we adjusted the charging voltage of the generator to output the demand impulse currents. Then, multiple lightning impulse currents were applied to the ZnO varistors. The time between the application of one group of impulse currents to a ZnO varistor block, and that of the next group of impulse currents was 30 minutes; with such long duration of time, we were able to return to the original conditions. The process of the impulses is shown in Figure 5.

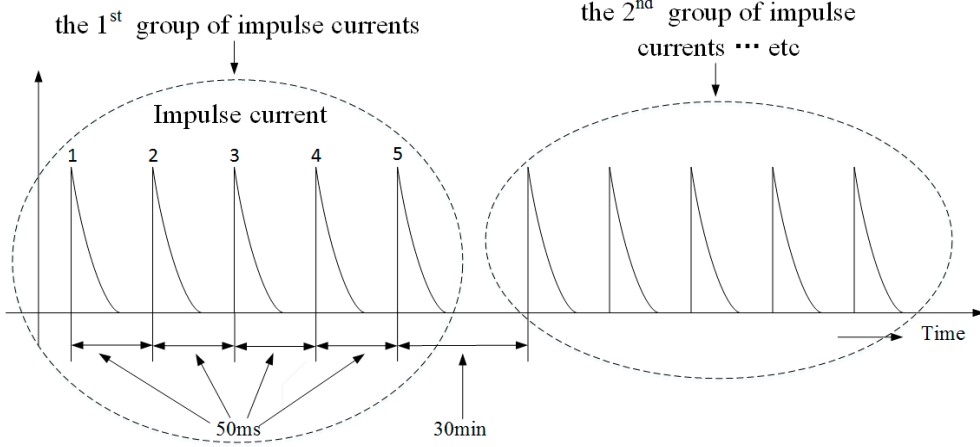

**Figure 5.** Groups of impulse currents applied to the ZnO varistors.

(3) We measured the surface temperatures, varistor voltage $U_{1mA}$, and leakage current $I_{ie}$ of the ZnO varistors after each impulse. Then, we checked the surface of the ZnO varistors for flashover or puncture and took photographs of the damaged ZnO varistors. When the change in amplitude of the $U_{1mA}$ reached beyond $\pm 10\%$ of the original, the $I_{ie}$ exceeded 20 μA, or direct damage occurred, the ZnO varistors were judged as having failed. Subsequently, the impulse test was ceased, and the data were recorded.

(4) We used scanning electron microscopy (SEM) and an X-ray diffractometer (XRD) on the ZnO varistor blocks after the impulse test in order to observe the microstructural changes of the ZnO varistors.

## 3. Results and Discussion

### 3.1. Macroscopic Properties

The average level change directions of the $U_{1mA}$ and $I_{ie}$ after the increases in the impulse groups for the ZnO varistors under multiple lightning impulse currents can be seen in Figure 6. It can be observed that the $U_{1mA}$ showed a trend of decreasing-stable-decreasing with the increase of impulse groups, and was observed to drop after the first group of impulses and the last group of impulses, with an average decline rate as high as 4%. The $I_{ie}$ all showed increasing trends with larger rising rates, with an average growth rate of 0.66 μA/a group. Following the 16 groups of impulses, it was observed that the $U_{1mA}$ fell sharply, with an average drop rate of more than 10% of the original $U_{1mA}$. This resulted in failures occurring in the ZnO varistors. Then, after another group of impulses, the ZnO varistor blocks became damaged.

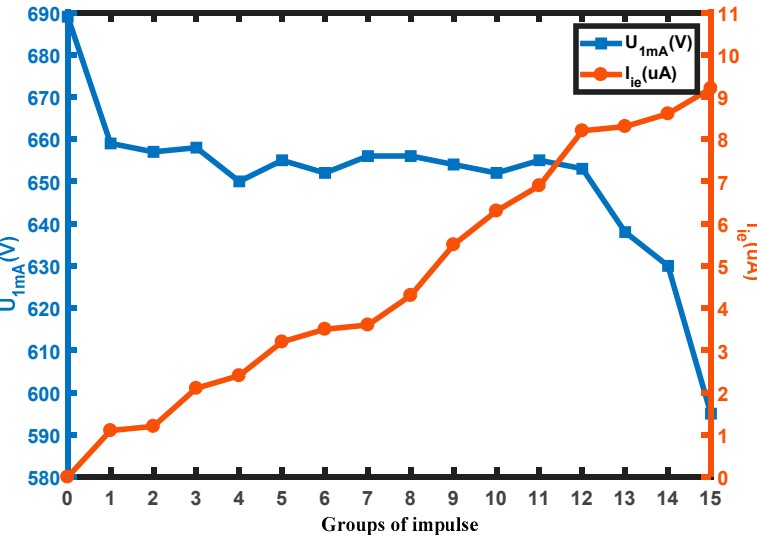

**Figure 6.** Varistor voltage $U_{1mA}$ and leakage current $I_{ie}$ variation diagrams of the ZnO varistors under multiple lightning impulse currents.

As shown in Table 1, under the five-pulse lightning currents, the ZnO varistor blocks A1 to A7 was able to withstand an average of 16 groups of impulses. When the $U_{1mA}$ had fallen one to two groups of impulses. Figure 7b shows the damage forms of the ZnO varistor block A2 during its failure, which presented as a side-corner cracking along with patch collapse [24]. As shown in Figure 7c, the damage forms of the ZnO varistor blocks A1 to A4 in the different codes were observed to be highly consistent.

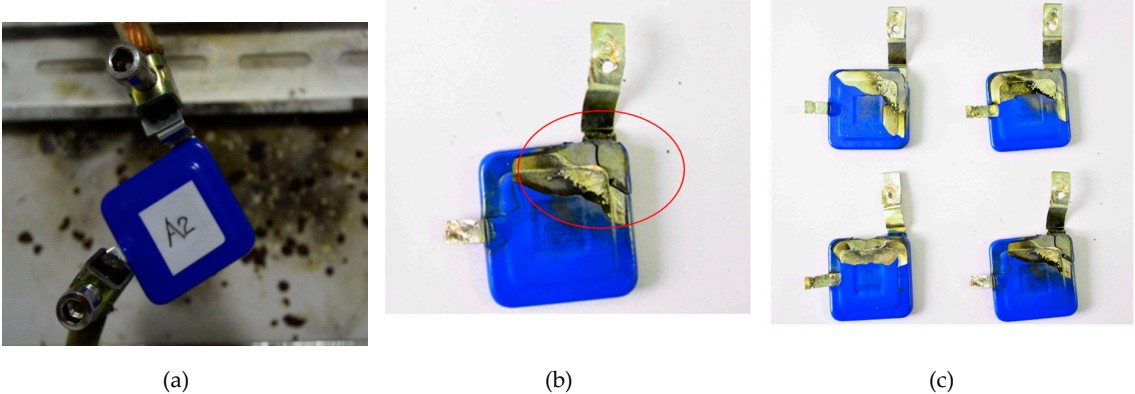

(a)                                          (b)                                          (c)

**Figure 7.** The appearance state diagram of the ZnO varistors before and after the multiple lightning impulse currents. (**a**) The state diagram of Sample A2 before impulses; (**b**) Damage state diagram of Sample A2; (**c**) Damage diagrams of Samples A1 to A4.

### 3.2. Microscopic Properties

Figure 8 details the SEM images of the ZnO varistors prior to the impulses, with a resolution of 10 μm. It can be seen from the figure that the internal structure of the ZnO varistor was mainly composed of four types of gray matter, marked as +1, +2, +3, and +4. Through this study's EDS analysis, it was determined that the main components of +1 in the white area were O, Zn, and Bi; those of +2 in the gray area were O and Zn; those of +3 at the white-gray junction were O, Zn, and Bi, and those of +4 in the deep black region were O and Zn. The proportion of elements in different regions is shown in Figure 8b. It can be seen that the main components of the ZnO varistor are zinc oxide and small amounts of Bi compounds. The gray area in the figure indicates the zinc oxide grain, and the white area indicates the rich-Bi phase around the zinc oxide grain. It was found that the cell distributions of the ZnO varistor were not uniform, and the rich-Bi phase was concentrated in a certain region. Map analysis at the +1 position of the ZnO varistors is shown in Figure 8c.

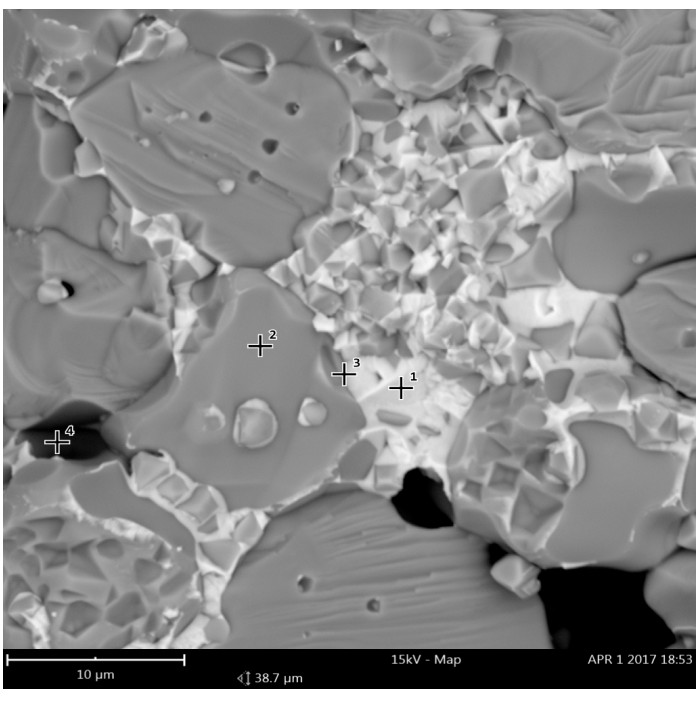

(a)

**Figure 8.** *Cont.*

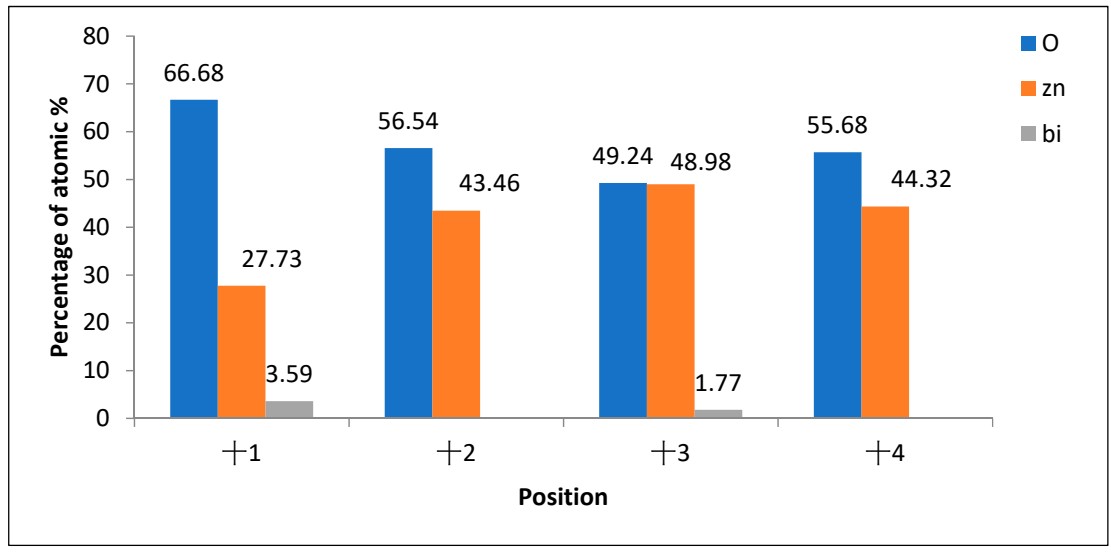

(b)

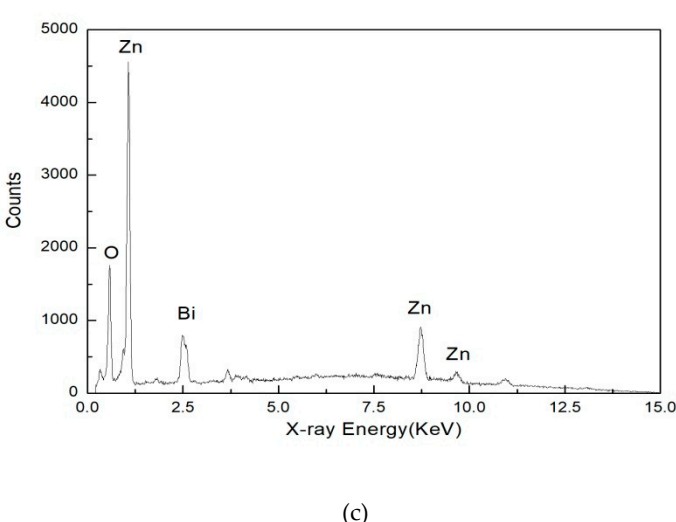

(c)

**Figure 8.** (**a**) SEM images of the ZnO varistors (resolution of 10 μm); (**b**) EDS analysis results of the ZnO varistors. (+1, +2, +3, +4); (**c**) Map analysis at the +1 position of the ZnO varistors.

Figure 9a,b shows the XRD diffraction pattern of the ZnO varistors, before and after the impulses. It can be seen in the figure that the crystal phase compositions mainly include four parts: zincite, syn; manganese oxide; bismuth oxide; and bismite, syn. Through the analysis of the figures, it was determined that the proportion of Bi in the crystal phases before the impulse test was 0.9%, 0.8%, and 0.6%, respectively. The proportion after the impulse test was 0.0%, 1.0%, and 1.3%, respectively. The proportion of Bi in the crystal phases has been converted.

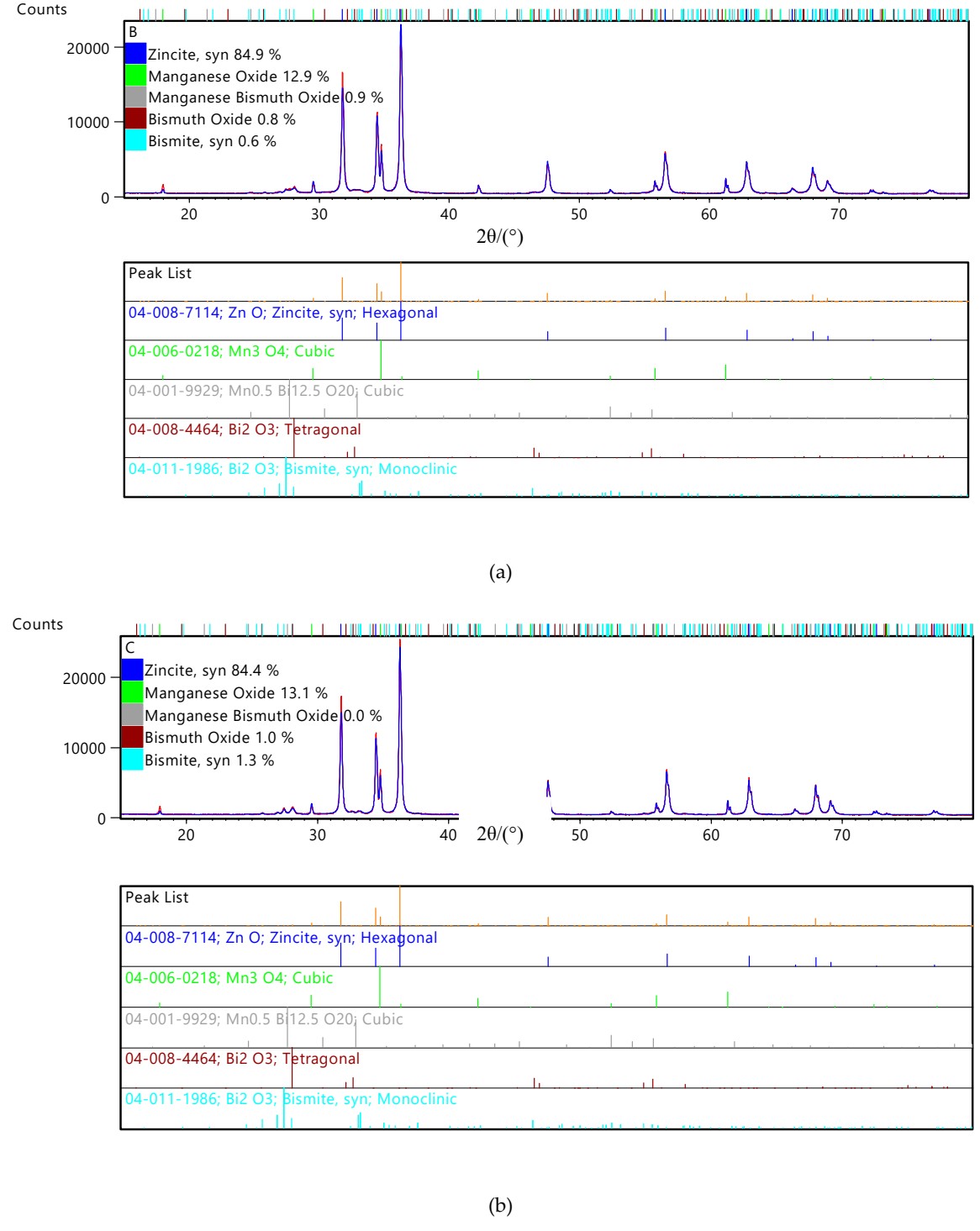

**Figure 9.** XRD diffraction results of the ZnO varistors before (**a**) and after (**b**) the impulses.

Figure 10a,b illustrate the SEM images of the ZnO varistors, before and after the impulses, with a resolution of 80 μm. It was observed that after the impulses, the grain sizes became smaller, while the white areas increased. The differences in the microstructures of the materials in the ZnO varistors indicate that the multiple lighting impulse currents led to the ZnO grains becoming smaller, along with the grain boundary growth.

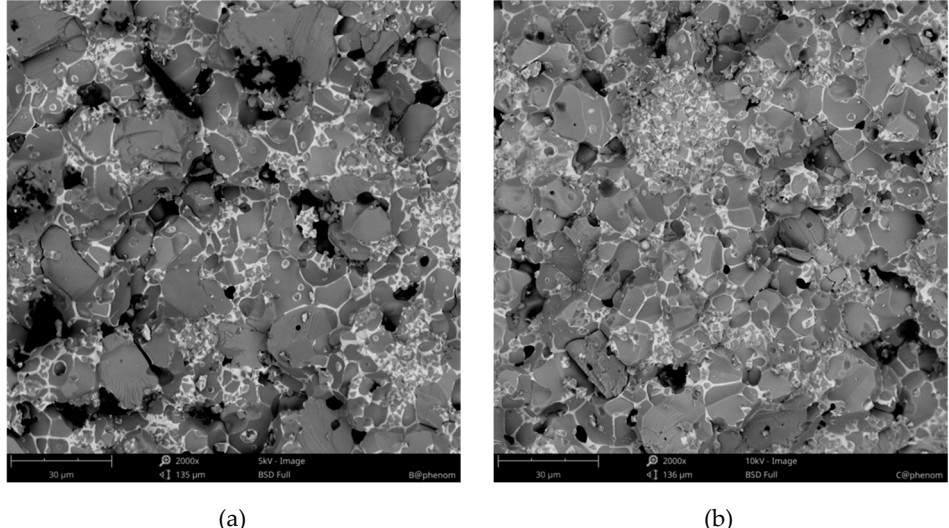

(a)                                                    (b)

**Figure 10.** SEM images of the ZnO varistors (resolution of 80 μm). (**a**) Before the impulses. (**b**) After the impulses.

### 3.3. Failure Mechanism

The ZnO varistors have the appearance of layers of insulation coating. When impulse current passes through the ZnO varistors, the pulse intervals are ms in length, and the internal heat of the ZnO varistors will be almost instantaneously concentrated, accompanied by a sharp rise in temperature. In terms of the energy convection and exchange, this process can be seen as an adiabatic temperature rise. At the time that the failures occurred in the ZnO varistors, the surface temperature had risen to over 200 °C, and the average temperature was 223 °C, as shown in Table 1. According to Equation (1), in the cases of the injected energy equivalent, the temperature increases of the ZnO varistors units with the intake of energy could be directly determined by the thermal physical property parameters $\rho$ and $C_p$. However, during the production process of the ZnO varistors, absolute material mixing uniformity could not be achieved. As can be seen from the SEM images of the ZnO varistor blocks, the microstructures show inhomogeneities of the internal material distribution, which were observed to have led to significant differences in the thermal physical properties of different parts of the same ZnO varistors. For example, inhomogeneities existed in the thermal physical properties of the ZnO varistor blocks. In this study, t was determined that the $\rho$ and $C_p$ differences in different parts led to different adiabatic temperature increases. Furthermore, the infrared imaging measurement results showed different temperature increases in different locations of the ZnO varistor blocks after absorbing the impulse energy. Figure 11 shows the temperature distribution diagram of the ZnO varistor blocks A1 and A4 following the impulses. It can be seen in the figure that there were considerable differences observed in the temperature increases in different parts. The energy at the local hot spot occurred too late to be passed around, which led to a large temperature gradient between the hot spot and the surrounding area. Thermal stress to the temperature gradient occurred in the interior of the ZnO varistor block. When the thermal stress reached a certain value, it caused burst damage to the ZnO varistor block [25].

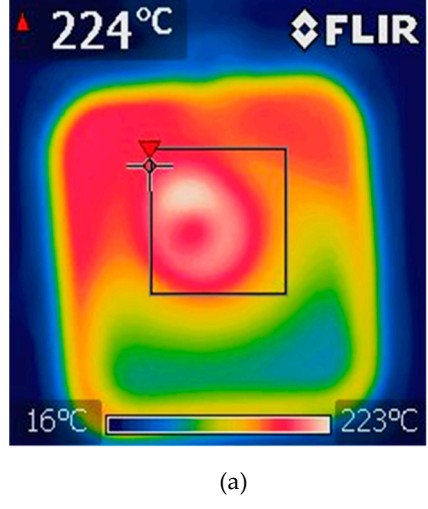 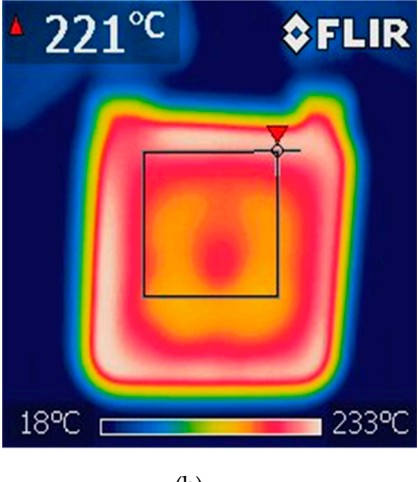

(a)　　　　　　　　　　　　　　　(b)

**Figure 11.** Temperature distribution map of the ZnO varistor after impulses. (**a**) Varistor block A1. (**b**) Varistor block A4.

The adiabatic temperature increases of the ZnO varistor units after the absorption of energy $W$ (unit: J) were as follows [25]:

$$\Delta T = \frac{\Delta W}{\Delta V \rho c_p} \tag{1}$$

In Figure 11, $\Delta V$ is the volume of the ZnO varistor units; $\rho$ is the proportion of the ZnO varistors, with an average of 5600 kg/m$^3$; and $C_p$ is the constant-pressure heat capacity of the ZnO varistor units, which was approximately 500 J/kg $^\circ$C at 20 $^\circ$C.

If the temperature increases of the adjacent two units were $\Delta T_1$ and $\Delta T_2$, respectively, then the thermal stress $f$ under the effects of the temperature gradient was as follows [25]:

$$f = \frac{Ea}{1 - \mu}(\Delta T_1 - \Delta T_2) \tag{2}$$

where $E$ is the elastic modulus; $a$ denotes the thermal expansion coefficient; and $\mu$ is the Poisson's ratio. When the thermal stress $f$ produced under the action of the temperature gradient exceeded the thermal stress threshold during the rupture failure of the ZnO varistors material, it could potentially lead to the destruction of the ZnO varistors.

The experimental phenomenon in which the varistor voltage presented a declining trend could be explained by the microstructure characteristics of the ZnO varistors. As shown in Figure 10, the SEM images confirmed that the interiors of the ZnO varistors were composed of many zinc-oxide grains, and the distances between various grains were fixed. Meanwhile, transverse and longitudinal capacitances were formed between the grains, which displayed a vertical distribution.

The microanalysis showed that the grain sizes of the ZnO varistors became smaller, while the area of the grain boundary layer increased, which led to the increases in the capacitance of the grain boundary layer as follows:

$$C = \frac{\varepsilon S}{4\pi kd} = \frac{Q}{V} \tag{3}$$

In Equation (3) [26], $C$ represents the capacitance; $\varepsilon$ is the dielectric constant; $S$ is the cross-sectional area; $k$ denotes the electrostatic force constant; $d$ is the anode-to-cathode distance; $Q$ represents the charge quantity; and $V$ is the voltage.

It can be seen from the above equation that, as the number of impulse groups increased, the capacitance of the grain boundary layer increased, while the varistor voltage gradually decreased.

In the ZnO varistors under the multiple lightning impulse current, the leakage current value gradually increased. This was due to the fact that after several impulse groups, the ZnO varistors

absorbed the impulse energy, and the temperatures rose. This sped up the rates of ion migration, resulting in the leakage current values displaying a growing trend.

## 4. Conclusions

In this study, experiments were conducted to investigate the microstructure and macroscopic properties of ZnO varistors under multiple lightning impulse currents.

(1) The macroscopic failure mode of ZnO varistors under multiple lightning impulses could be described as the rapid deterioration of the electrical parameters with the increase of the number of impulse groups before destruction occurred by side-corner bursting. Under a five-pulse lightning current, the ZnO varistors were able to withstand an average of 16 groups of impulses.

(2) The microstructural examination indicated that, after the multiple lightning strokes, the proportion of Bi in the crystal phases was converted, the grain size of the ZnO varistors became smaller, and the white intergranular phase (Bi-rich grain boundary layer) increased significantly.

(3) The ZnO varistors undergoing multiple lightning impulse currents presented adiabatic temperature increases. It was observed that, due to the unevenness of the material, the ZnO varistors displayed local temperature increases after absorbing impulse heat. The failure mechanism was thermal damage and grain boundary structure damage caused by the temperature gradient thermal stress generated by multiple lightning currents.

**Author Contributions:** Data curation, C.L.; Investigation, H.X.; Methodology, D.L. and S.Y.; Writing—original draft, C.Z. and P.L.

**Funding:** This research was funded by the National Nature Science Foundation of China, NSFC Grants No. 61671248, Open Foundation of Jiangsu Key Laboratory of Meteorological Observation and Information Processing, KDXS1603, and A Project Funded by the Priority Academic Program Development of Jiangsu Higher Education Institutions (PAPD) and The APC was funded by Heilongjiang Provincial Meteorological Bureau Scientific Research Project, HQ2015005.

**Acknowledgments:** The authors are thankful to the anonymous reviewers for their valuable comments.

**Conflicts of Interest:** The authors declare no conflict of interest.

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
