# Peer review of "An Experimental Study of the Failure Mode of ZnO Varistors Under Multiple Lightning Strokes"

_electronics, doi:10.3390/electronics8020172_

Round 1

Reviewer 1 Report

The authors have conducted the experiments to investigate the properties of ZnO varistors under multi-lightning impulse current. The whole manuscript needs major revision for language. The experimental section needs further clarification and language modification for better understanding. Please improve the introduction, include more references.

Author Response

Point 1: The whole manuscript needs major revision for language.

Response 1:

The whole manuscript has been greatly revised in language. The manuscript has been Entrusted MDPI for English Pre Editing editing.

Point 2: The experimental section needs further clarification and language modification for better understanding.

Response 2:

I have clarificated flowchart of the experimental procedure. Fig.4(a) is before the clarification, and the Fig.4(b) on the right is the clarificated flowchart. The flowchart of the experimental procedure clarificated is more relevant to the actual test process. At the same time, language has been modified.

 Fig.4(a ) Fig.4(b)

Point 3: Please improve the introduction, include more references.

Response 3:

I have improved the introduction, and the supplementary content is as follows:

(1)When a low-voltage distribution line is struck by lightning, the lightning current flowing through the surge protective device generates heat in the ZnO varistor due to power loss, causing the body to heat up. When a multi-pulse lightning current and single-pulse lightning current flow through a ZnO varistor, there is a huge difference in duration and energy absorption, which inevitably leads to a difference in the temperature increase of the ZnO varistor and the final electrical performance parameters.

(2)Qingheng Chen et al. [20] used the simulation network model to analyze the current, temperature, and thermal stress in zinc oxide varistors. The results showed that reducing the average size of the ZnO grains can significantly reduce the temperature difference inside the ZnO varistor. Thermal stress increases the impact energy absorption capacity of zinc oxide varistors.

(3)Pengfei Li et al. [21] analyzed the damage form of metal oxide under multiple lightning impulses, and found that it was mainly edge cracking.

(4)Toshihiro Tsuboi et al. [22] believed that damage could occur in the internal components of ZnO when subjected to multiple lightning impulses, resulting in the failure of the ZnO varistor, which provided a theoretical basis for the modeling of this study.

Reviewer 2 Report

This paper presents an experimental study of the failure mode of ZnO varistors after their exposure to multiple lightning strokes. The paper is well written and structured and the idea is adeptly explained. I believe the manuscript would be beneficial for the wide readers of electronics, hence, I want to accept the manuscript for publication after minor revision of the comments.

1- Provide a bibliographic reference to the facts reported in line 51-53.

2- References [9-10] and [13-14} should be written as [9,10] and [13,14], respectively. Have a look at the rest of the references as well and write them as per the formal electronics' referencing format.

3- Provide the source bibliographic references to the equations 1, 2, and 3 for readers' understanding.

Author Response

Point 1: 1- Provide a bibliographic reference to the facts reported in line 51-53.

Response 1:

A bibliographic reference to the facts reported in line 51-53 :

Darveniza M.;Tumma L R. Multipulse lightning currents and metal-oxide arresters. IEEE Transactions on Power Delivery, 1997, 12(3), 1168–1175.

Point 2: References [9-10] and [13-14] should be written as [9,10] and [13,14], respectively. Have a look at the rest of the references as well and write them as per the formal electronics' referencing format.

Response 2:

All the references have been written as per the formal electronics' referencing format.

Point 3: Provide the source bibliographic references to the equations 1, 2, and 3 for readers' understanding.

Response 3:

The source bibliographic references to the equations 1, 2:

Wu W.;He J. Impact damage principle of metal oxide valve plates. In: Properties and Applications of Nonlinear Metal Oxide, Varistors, 5nd ed. ;Tsinghua University Club; Beijing, China, 1998, pp.147–149.

The source bibliographic references to the equation 3:

Zou R. Discussion on Dielectric Constants in Secondary School Physics Textbooks. Technical Physics Teaching, 2000,4,22-23.

Round 2

Reviewer 1 Report

Thank you for the revisions.